# SNP-by-CpG Site Interactions in *ABCA7* Are Associated with Cognition in Older African Americans

**DOI:** 10.3390/genes13112150

**Published:** 2022-11-18

**Authors:** Dima L. Chaar, Kim Nguyen, Yi-Zhe Wang, Scott M. Ratliff, Thomas H. Mosley, Sharon L. R. Kardia, Jennifer A. Smith, Wei Zhao

**Affiliations:** 1Department of Epidemiology, School of Public Health, University of Michigan, Ann Arbor, MI 48109, USA; 2Memory Impairment and Neurodegenerative Dementia (MIND) Center, University of Mississippi Medical Center, Jackson, MI 39216, USA; 3Survey Research Center, Institute for Social Research, University of Michigan, Ann Arbor, MI 48104, USA

**Keywords:** general cognitive function, cognition, *ABCA7*, African American, Alzheimer’s disease

## Abstract

SNPs in *ABCA7* confer the largest genetic risk for Alzheimer’s Disease (AD) in African Americans (AA) after *APOE* ε4. However, the relationship between *ABCA7* and cognitive function has not been thoroughly examined. We investigated the effects of five known AD risk SNPs and 72 CpGs in *ABCA7*, as well as their interactions, on general cognitive function (cognition) in 634 older AA without dementia from Genetic Epidemiology Network of Arteriopathy (GENOA). Using linear mixed models, no SNP or CpG was associated with cognition after multiple testing correction, but five CpGs were nominally associated (*p* < 0.05). Four SNP-by-CpG interactions were associated with cognition (FDR q < 0.1). Contrast tests show that methylation is associated with cognition in some genotype groups (*p* < 0.05): a 1% increase at cg00135882 and cg22271697 is associated with a 0.68 SD decrease and 0.14 SD increase in cognition for those with the rs3764647 GG/AG (*p* = 0.004) and AA (*p* = 2 × 10^−4^) genotypes, respectively. In addition, a 1% increase at cg06169110 and cg17316918 is associated with a 0.37 SD decrease (*p* = 2 × 10^−4^) and 0.33 SD increase (*p* = 0.004), respectively, in cognition for those with the rs115550680 GG/AG genotype. While AD risk SNPs in *ABCA7* were not associated with cognition in this sample, some have interactions with proximal methylation on cognition.

## 1. Introduction

Alzheimer’s disease (AD) is a progressive neurodegenerative disease characterized by the dysregulation of the amyloid-β (Aβ) pathway, leading to Aβ plaques [1] and the aggregation of tau tangles [2]. AD accounts for 60–80% of dementia cases in the elderly [3,4,5]. Approximately 6.2 million Americans age 65 and older are living with AD, and this estimate is projected to rise to 13.8 million by 2060 [3]. AD risk differs by race, with African Americans (AA) twice as likely to develop AD compared to European Americans (EA) [6]. Because this health disparity places a greater burden of personal and medical care on AA, it is crucial to better understand AD and its development in this population.

AD is a multifactorial disease that is likely influenced by interactions between genetic, environmental, and epigenetic factors, along with age-related neurodegeneration [7]. In addition to age, genetic variants in the apolipoprotein E (*APOE*) gene are the largest risk factor for AD in AA [8], with one copy of the *APOE* ε4 allele increasing AD risk 3–5-fold [9,10,11]. *ABCA7* is the second largest genetic risk factor for AD in AA, with genetic variants increasing AD risk by 70–80% [8]. The *ABCA7* gene encodes the ATP-binding cassette (ABC) transporter A7, which regulates homeostasis of phospholipids and cholesterol in the central nervous system and peripheral tissues [12,13,14]. This gene is mostly expressed in the brain, spleen, lungs, and adrenal gland [15]. Studies have suggested that mutations in *ABCA7* are associated with AD susceptibility through the dysregulation of lipid metabolism, which facilitates Aβ clearance [13,14].

Though *ABCA7* is a risk locus for AD in both EA and AA, the specific risk variants differ across groups [16]. In EA, three *ABCA7* SNPs, rs3764650, rs3752246, and rs4147929, are associated with AD. They represent two independent signals, as rs3752246 and rs4147929 are in nearly complete linkage disequilibrium (LD) in EA. Although rs3764650 shows the strongest association with AD in EA, it is only nominally associated in AA [16,17]. In AA, two additional *ABCA7* SNPs, rs3764647 and rs3752239, have stronger associations with AD [17], with rs3764647 being in the same LD block as rs3764650 in AA. Interestingly, another independent SNP in *ABCA7*, rs115550680, which is monomorphic in EA, is strongly associated with AD in AA. In particular, the G allele of rs115550680 confers an AD risk comparable to *APOE* ε4 (OR = 1.79) in AA [8].

Epigenetic modifications, such as DNA methylation, are potential molecular mechanisms that can modulate the effect of genetic risk factors [18]. When methylation sites (CpGs) are clustered together as a CpG island (CGI), it often serves as a hub for gene expression regulation. CGIs in the promoter region usually suppress transcription, whereas CGIs in the intragenic region can interact with multiple regulatory elements and have a variety of impacts on gene expression (e.g., influencing mRNA isoforms or promoting enhancer function) [19]. Given the regulatory role of DNA methylation on gene expression, there has been a growing interest in understanding the extent to which DNA methylation contributes to AD risk [20,21,22,23,24]. In particular, recent studies on post-mortem brain tissue found evidence of association between DNA methylation in *ABCA7* and both AD and AD-related pathologies, including Aβ load and tau tangle density [21,22]. This evidence suggests that methylation in *ABCA7* has a non-trivial functional role which is worthy of further investigation.

Although the relationships between AD and *ABCA7* SNPs are well-characterized, there are limited studies on the association between genetic variation in *ABCA7* and measures of cognitive function and/or cognitive decline prior to the development of dementia. An imaging study showed that *ABCA7* SNPs were associated with amyloidosis among cognitively healthy individuals and those with mild cognitive impairment, but not among those with AD, suggesting an early effect of *ABCA7* on cognition and cognitive decline [25]. A few studies in EA found inconsistent results regarding the effect of *ABCA7* SNPs on cognition, with associations varying by sex, *APOE* status, and disease progression [26]. For example, in healthy older adults, a longitudinal study found an association between rs3764650 and cognitive decline, but only in females [27]. Additionally, interactions between the *APOE* ε4 allele and SNPs rs3764650 and rs3752246 were associated with three cognitive factor scores related to verbal learning and memory, working memory, and intermediate memory, in a genotype-dependent manner: in the absence of *ABCA7* minor alleles, each additional ε4 allele was associated with lower memory scores; and conversely, in the presence of *ABCA7* minor alleles, each additional ε4 allele was associated with better memory scores [28]. Lastly, rs3764650 was significantly associated with increased rates of memory decline among individuals with mild cognitive impairment or AD [29].

To our knowledge, no study has investigated the relationship between *ABCA7* genetic variation and cognition in cognitively healthy AA. Further, few studies have examined the relationship between DNA methylation in *ABCA7* and/or its interaction with genetic variants on general cognitive function. In this study, we investigate whether previously identified risk SNPs (referred to as sentinel SNPs) in *ABCA7*, DNA methylation in *ABCA7*, and their interactions are associated with general cognitive function in older AA without dementia. In order to better understand the functional consequences of these risk factors at the molecular level, we also evaluated whether identified epigenetic or genetic risk factors are associated with transcript level *ABCA7* gene expression in transformed B-lymphocytes from the same cohort. A thorough investigation of the relationship between these multi-omic layers and later-life cognition can help characterize the underlying genetic architecture of cognition in older adulthood, prior to dementia onset. This may allow the identification of targets for intervention and treatment, especially in populations that are most at risk [30].

## 2. Materials and Methods

### 2.1. Sample

The Genetic Epidemiology Network of Arteriopathy (GENOA) study is a community-based longitudinal study aimed at examining the genetic effects of hypertension and related target organ damage [31]. European American (EA) and African American (AA) hypertensive sibships were recruited if at least two siblings were clinically diagnosed with hypertension before age 60. All other siblings were invited to participate, regardless of hypertension status. Exclusion criteria included secondary hypertension, alcoholism or drug abuse, pregnancy, insulin-dependent diabetes mellitus, active malignancy, or serum creatinine levels > 2.5mg/dL. In Phase I (1996–2001), 1854 AA participants (Jackson, MS) and 1583 EA participants (Rochester, MN) were recruited [31]. In Phase II (2000–2004), 1482 AA participants and 1239 EA participants were successfully followed up, and their potential target organ damage from hypertension was measured. Demographics, medical history, clinical characteristics, information on medication use, and blood samples were collected in each phase. Methylation levels were measured only in AA participants using blood samples collected in Phases I and II. In an ancillary study (2001–2006), 1010 AA and 967 EA GENOA participants underwent a battery of established neurocognitive tests to assess several measures of cognitive function, including learning, memory, attention, concentration, and language. Written informed consent was obtained from all participants, and approval was granted by participating institutional review boards (University of Michigan, University of Mississippi Medical Center, and Mayo Clinic).

A total of 850 AA participants had non-missing genetic and demographic data. Since participants with a history of stroke or dementia may have had changes in general cognitive function that differed from non-pathological cognitive aging, we excluded those who had a history of stroke (*n* = 43) and/or preliminary evidence of dementia as indicated by a score of <24 on the Mini-Mental State Examination (MMSE) (*n* = 76) [32]. We also excluded participants younger than age 45 (*n* = 16). A total of 634, 494, and 429 participants were available for SNP, methylation, and gene expression analyses, respectively (Appendix A).

### 2.2. Measures

#### 2.2.1. General Cognitive Function

General cognitive function was calculated using five neurocognitive measures evaluated at Phase II [32,33]:The Weschler Adult Intelligence Scale-Revised: Digit Symbol Substitution Test (DSST) measured complex visual attention, sustained and focused concentration, response speed, and visuomotor coordination. The DSST is related to the executive function of working memory in cognition [34]. In this test, participants matched symbols to numbers according to a key located at the top of the page. The DSST score comprised the number of symbols correctly matched within 90 s.The Controlled Oral Word Association Test (COWA-FAS) tested for verbal fluency (phonetic association) and language. This required participants to generate as many words as possible that start with F, A, and S in 1 min. The score consisted of the total number of admissible words generated.The Rey Auditory Verbal Learning Test (RAVLT) measured delayed recall, relating to the cognitive functions of new learning, immediate memory span, and vulnerability to learning interference, and recognition memory. Scores were determined by the number of words recalled after a 30-min delay. Scores ranged from 0 to 15.The Stroop Color–Word Test (SCWT) assessed concentration effectiveness by requiring participants to state the color of a word, rather than the word written. The score sums the number of color words that were correctly stated in 45 s. Specifically, the ability to shift perceptual sets in response to novel stimuli was tested.The Trail Making Test A (TMTA) evaluated visual conceptual tracking, as participants are required to connect a set of 25 circles quickly and accurately. TMTA provided information on the cognitive functions of visual search, scanning, processing speed, and executive functions. The TMTA score was measured as the amount of time (seconds) the participants took to complete the task. The maximum time allowed was 240 s. Prior to analysis, TMTA scores were natural log-transformed and recoded so that higher scores indicated better cognitive function.

General cognitive function, a measure of overall cognitive performance, can be quantified as a summary measure of cognitive tests in multiple cognitive domains [35]. In this study, general cognitive function was calculated as the first unrotated principal component (FUPC) from a principal component analysis (PCA) of the five neurocognitive measures in the full sample (*n* = 634). The FUPC accounted for 53% of the total variance in the neurocognitive measures, and loading values of the five measures ranged from 0.52 to 0.87.

#### 2.2.2. Demographic Data

Age was assessed at the time of cognitive testing. Educational attainment, measured at Phase II, was categorized into a three-level variable: (1) less than high school degree (reference group), (2) high school degree or GED, and (3) at least some college. Smoking has been shown to have a substantial impact on the epigenome [36], so we used smoking data concurrent with DNA methylation measures (Phase I). Participants were categorized as current, former, or never smokers (reference group).

#### 2.2.3. Genetic Data

Blood samples were genotyped using the Affymetrix^®^ Genome-Wide Human SNP Array 6.0 or the Illumina 1M Duo. Samples and SNPs with a call rate <95%, samples with mismatched sex, and duplicate samples were removed. Genotypes were imputed using the 1000 Genomes Project phase I integrated variant set (v.3) (Hg19, released in March 2012). Of the six SNPs of interest identified from the existing literature (rs3764647, rs3764650, rs115550680, rs3752246, rs3752239, and rs4147929), five had high imputation quality (r^2^ > 0.7), and one (rs3752239) was excluded due to low imputation quality (r^2^ = 0.49). SNPs were coded as the dosage of the corresponding AD risk allele, as specified in the previous literature. Genetic principal components were calculated from genotyped SNPs and included in regression models to control for population stratification. In order to evaluate confounding and/or effect modification by *APOE* isoforms known to influence dementia risk, we measured rs7412 (to capture the *APOE* ε2 allele) and rs429359 (to capture the *APOE* ε4 allele) using a TaqMan assay and ABI Prism^©^ Sequence Detection (Applied Biosystems, Foster City, CA, USA) in 1544 participants. Participants were classified as having 0, 1, or 2 copies of ε2 (represented by the rs7412 T allele) and/or ε4 (represented by the rs429359 C allele).

#### 2.2.4. DNA Methylation Data

Genomic data was extracted from stored peripheral blood leukocytes from 1106 AA participants from Phase I and 304 AA participants from Phase II using the AutoGen FlexStar (AutoGen, Holliston, MA, USA). Bisulfite conversion was performed with the EZ DNA Methylation Kit (Zymo Research, Irvine, CA, USA), and methylation was measured using the Illumina HumanMethylationEPIC BeadChip. The raw intensity data were visualized using the *shinyMethyl* R package [37] to identify sex mismatches and outliers, which were removed. Samples with incomplete bisulfite conversion were identified using Qcinfo in the *Enmix* R package [38], and then removed. Background correction and dye-bias normalization were performed using Noob in the *Minfi* R package [39,40]. We also checked sample identity using the 59 SNP probes on the EPIC chip, and mismatched samples were removed. Probe-type bias was adjusted using the Regression on Correlated Probes (RCP) method [41]. Probes with detection *p*-value < 10^−16^ were considered successfully detected, and probes and samples with a detection rate of <10% were removed [42]. After quality control, a total of 1396 samples (*n* = 1100 from Phase I and *n* = 294 from Phase II) and 857,121 CpG sites were available for analyses. For this analysis, all methylation data were collected from Phase I samples.

We selected all CpG sites within 5kb of the *ABCA7* gene (a total of 72 CpG sites within the *ABCA7* region: chr19, 1040102–1065570, and hg19). We used Illumina annotation [43] to characterize each CpG site as being in a promoter region and/or CGI, CGI shore, or CGI shelf. White blood cell proportions for CD8+ T lymphocytes, CD4+ T lymphocytes, natural killer cells, B cells, monocytes, and granulocytes were estimated using the Houseman method [44]. For each CpG site prior to analysis, the methylation beta-value [45,46] was multiplied by 100 to approximate the percent of methylation at that site. Methylation beta-values were pre-adjusted for batch effects (sample plate, row, and column) and white blood cell proportions using linear mixed modeling, and the resulting residuals were added to the mean values.

#### 2.2.5. Gene Expression Data

Gene expression levels in transformed B-lymphocyte cell lines from blood samples taken primarily at GENOA Phase II were measured using the Affymetrix Human Transcriptome Array 2.0. The Affymetrix Expression Console was used for quality control, and all array images passed visual inspection. Affymetrix Power Tool software was used to process raw intensity data [47]. We normalized Affymetrix CEL files using the Robust Multichip Average (RMA) algorithm, including background correction, quantile normalization, log_2_-transformation, and probe set summarization [48]. Linearity was also maintained using GC correction (GCCN), signal space transformation (SST), and gain lock (value = 0.75). We used the Brainarray custom CDF [49] version 19 to map the probes to genes, specifically removing probes with non-unique matching cDNA/EST sequences that could be assigned to more than one gene cluster. As a result, the gene expression data processed through the custom CDF are expected to be free of mappability issues; however, alignment bias may still exist due to genetic variation, errors in reference genome, and other complications [50]. After mapping, Combat was used to remove batch effects [51].

### 2.3. Statistical Analysis

#### 2.3.1. Genetic Analysis

We first calculated Pearson correlations between sentinel SNPs. Next, the association between *ABCA7* sentinel SNPs and general cognitive function was analyzed using linear mixed models with random effects to adjust for relatedness. Model 1 adjusted for age at cognitive testing, sex, and the first four genetic principal components (PC1-4), with family as a random effect to account for sibships. Model 2 additionally adjusted for educational attainment. Model 3 further adjusted for *APOE* ε2 and ε4. For any SNPs that were significantly associated with general cognitive function, we further examined the association between those SNPs and each of the five neurocognitive measures to identify the domain(s) that most strongly drive the association. Since prior studies have suggested that the effect of *ABCA7* SNPs may vary by sex, education, and/or *APOE* status, we also assessed the interaction between the sentinel SNPs and sex, education, or *APOE* (ε2 and ε4) on cognitive outcomes.

#### 2.3.2. Epigenetic Analysis

Pearson correlations were calculated for all 72 CpG sites. Next, linear mixed models were used to test the associations between each of the 72 CpG sites and general cognitive function. Model 1 adjusted for age at cognitive testing, sex, four genetic principal components, age difference between methylation and cognition measurements, smoking status, and family as a random effect to account for sibships. Model 2 additionally adjusted for educational attainment, and Model 3 further adjusted for *APOE* ε2 and ε4. The *coMET* package in R was used to create a regional plot to visualize association *p*-values, correlations, and Ensembl genes [52]. *BioRender* was used to annotate and format the figure [53]. For any CpGs that were significantly associated with general cognitive function, we further examined the association between those CpGs and each of the five neurocognitive measures in order to identify the domain(s) that most strongly drive the association.

#### 2.3.3. Genetic-Epigenetic Interaction Analysis

Next, we examined the interaction between each CpG site and sentinel *ABCA7* SNPs in association with general cognitive function. In this analysis, we adjusted for age at cognitive testing, sex, four genetic principal components, age difference between methylation and cognition measurements, smoking status, and *APOE* ε2 and ε4, with family as a random effect to account for sibships (Model 4). Models 1–4, which were used to assess genetic, epigenetic, and genetic–epigenetic interaction associations with general cognitive function are shown in Appendix A. To improve interpretability, we mean-centered methylation so that the β estimates from the regression models reflect the effect sizes for those with average methylation in the population. For any identified significant interaction, we stratified the genotypes by number of risk alleles (0, 1, or 2 risk alleles) and conducted contrast tests using the Emtrends function in the *Emmeans* package in R [54] to obtain the effect size of the CpG associated with general cognitive function in each genotype group. Minor homozygote genotype groups that made up <5% of the sample size were grouped with heterozygous genotype groups to increase power as appropriate. Plots of SNP-by-CpG interactions on general cognitive function were generated using the *effects* [55] and *ggplot2* [56] packages in R. Any identified SNP-by-CpG interactions significantly associated with general cognitive function were also tested for association with each of the five neurocognitive measures.

As a sensitivity analysis for significant interactions (FDR q < 0.1), we tested the association after excluding outlying CpG values that were more than four standard deviations from the mean (Model 4). We then assessed whether the SNP-by-CpG interactions (FDR q < 0.1) were driven by potential SNP-CpG correlations by testing the association between each SNP and its corresponding CpG, adjusting for age at methylation measurement, sex, and the first four genetic principal components, with family as a random effect. If the SNP and CpG were associated at *p* < 0.05, we adjusted out the effect of the SNP from the CpG site and re-tested the interaction (Model 4).

#### 2.3.4. Gene Expression Analysis

Among the 494 participants with methylation and genetic data, 429 participants also had gene expression data. Appendix A presents a graphical depiction of *ABCA7* transcripts observed in the Genotype Tissue Expression (GTEx) project [57], which assesses gene expression levels in a variety of cell types. A total of 17 transcripts, along with a measure of overall *ABCA7* gene expression, were available for analysis in our study. For SNPs, CpGs, or interactions that were significantly associated with general cognitive function, we assessed their association with *ABCA7* gene-level and transcript-level expression (Model 5) using linear mixed models. Model 5 adjusted for age at which gene expression data was generated (age at blood draw), sex, first four genetic principal components, and family as a random effect. For models that included CpG sites, Model 5 also included the age difference between methylation and gene expression measurements. Similarly, for any significant interaction effects, contrast tests were conducted to obtain the effect size in each genotype group. Minor homozygote genotype groups (<5% sample size) were grouped with heterozygous genotype groups to increase power as appropriate.

We next evaluated whether the identified CpG sites within the *ABCA7* region correlate with gene expression of *ABCA7* and/or nearby genes in an external public database with multiple cells/tissues. For this, we used *cis-* expression quantitative trait methylation (*cis*-eQTM) results from peripheral blood mononuclear cells (PBMCs) and three specific white blood cell types (CD4 + T lymphocytes, monocytes, and neutrophils) in the iMETHYL database [58,59], which integrated genotype, methylation, and gene expression data from 102 individuals. We also examined gene expression levels of *ABCA7* in different cell types available from the Genotype Tissue Expression (GTEx) project [57].

#### 2.3.5. Multiple Testing Correction

All statistical analyses were conducted using R (Version 3.6) [60]. For genetic analysis, the Bonferroni corrected *p*-value cut-off (*p* < 0.05/5) was used to claim significance. For all other analyses, false discovery rate (FDR) correction was applied to each model, and FDR q < 0.1 was considered significant. Since the SNPs, CpG sites, and transcripts in *ABCA7* were all correlated, applying stringent multiple testing corrections might have been too conservative; thus, any nominal associations were also noted.

## 3. Results

### 3.1. Sample Characteristics

The sample included 634 AA without dementia (Table 1). Overall, participant age ranged from 45 to 85 years (mean = 63.3 years), and the mean age difference between Phase I methylation and cognitive measurements was 6.0 years (SD = 1.3). More than half of participants (74.9%) were female, and 47.3% had at least some college education. General cognitive function was normally distributed. Mean RAVLT score was 7.1 (SD = 3.3) words recalled, mean DSST score was 34.4 (SD = 12.6) symbols, mean COWA-FAS score was 29.7 (SD = 11.6) words, mean SCWT score was 22.5 (SD = 9.8) items, and mean TMTA score was 61.6 (SD = 32.0) seconds to completion.

### 3.2. Correlation among Six Cognitive Outcomes

Pearson correlations (r) among the six cognitive outcomes (general cognitive function and the five individual neurocognitive measures) are shown in Appendix A. The five neurocognitive measures were moderately correlated (Pearson r ranged from 0.24 to 0.66), with the highest correlation between DSST and TMTA (r = 0.66, *p* < 0.001).

### 3.3. Correlation among ABCA7 SNPs 

Pearson correlations among the five sentinel *ABCA7* SNPs are shown in Appendix A. Rs3764647 was strongly correlated with rs3764650 (r = 0.84, *p* < 0.001), and rs3752246 was highly correlated with rs4147929 (r = 0.96, *p* < 0.001). The other sentinel SNP pairs were only weakly correlated or uncorrelated. 

### 3.4. Genetic Associations

In Models 1 and 2, there were no *ABCA7* SNPs that met the nominal significance threshold (*p* < 0.05, Appendix A). Although *APOE* was not part of the primary analysis, *APOE* ε2 and ε4 were analyzed separately as exposures in Models 1 and 2. *APOE* ε4 was associated with general cognitive function in both models in the expected direction (higher dosage of ε4 was associated with lower cognitive function). After adjusting for educational attainment and *APOE ε2* and *ε4* in Model 3, sentinel SNPs remained unassociated with general cognitive function. There were no observed nominal or significant interactions between SNPs and sex, *APOE* isoforms, or educational attainment on general cognitive function. 

### 3.5. Epigenetic Associations

Among the 72 CpG sites examined, six were nominally associated with general cognitive function in at least one of the three Models (Appendix A). After adjusting for educational attainment and *APOE* ε2 and ε4 (Model 3), five CpGs (cg22271697, cg00874873, cg11714200, cg26264438 and cg12082025) in the *ABCA7* region were nominally associated with general cognitive function. Figure 1 illustrates the regional plot of association *p*-values of the 72 CpGs in the *ABCA7* region with general cognitive function according to the chromosomal positions of CpG sites, as well as the correlations between the CpGs (Model 3).

### 3.6. Genetic-Epigenetic Interactions

Since rs3764647 and rs3764650, as well as rs4147929 and rs3752246, are highly correlated with each other (Appendix A), we removed one SNP from each pair. We selected rs3764647 because it had stronger evidence of association with AD in AA than rs3764650 [8]. We selected rs3752246 because it was a missense variant and more likely to have a functional effect than rs4147929, which is intronic [61]. Thus, we analyzed three independent risk SNPs (|r| < 0.60) in the interaction analysis. We assessed the interaction between each of the three independent sentinel SNPs (rs3764647, rs115550680, and rs3752246) and 72 CpG sites on general cognitive function, and identified four significant SNP-by-CpG interactions (FDR q < 0.1) that were associated with general cognitive function (Table 2): rs3764647*cg00135882 (*p* = 1.46 × 10^−4^), rs3764647*cg22271697 (*p* = 5.77 × 10^−4^), rs115550680*cg06169110 (*p* = 2.18 × 10^−4^), and rs115550680*cg17316918 (*p* = 4.84 × 10^−4^). The two SNPs and four CpGs that were involved in the four significant SNP-by-CpG interactions are shown in Figure 1 to highlight their positions with respect to neighboring genes, regulatory elements, and CGIs in the *ABCA7* region. All interactions with at least nominal significance are shown in Appendix A. Notably, an additional seven CpG sites had nominally significant interactions with rs115550680, and one additional site had a nominally significant interaction with rs3764647. In Appendix A, we present Pearson correlations among the *ABCA7* CpG sites that were nominally associated with general cognitive function (Appendix A) and/or were involved in an FDR-significant SNP-by-CpG interaction (Table 2). The majority of these CpGs were weakly correlated or uncorrelated.

For interactions with FDR q < 0.1, we performed contrast tests to estimate the effect size of the specific CpG site per genotype group. In all four cases, the minor homozygote genotype group had a small frequency (<5% of the sample size); thus, we combined them with the corresponding heterozygote genotype group. Contrast tests showed that methylation is associated with general cognitive function in some genotype groups, but not others (*p* < 0.05; Table 3 and Figure 2).

Rs3764647 had significant interactions with two CpGs (cg00135882 and cg22271697). For those with the risk genotype (GG/AG), a 1% increase at cg00135882 was associated with a 0.68 SD decrease in general cognitive function (*p* = 0.004, Figure 2A); whereas for those with the AA genotype, a 1% increase at cg22271697 was associated with a 0.14 SD increase in general cognitive function (*p* = 2.00 × 10^−4^, Figure 2B). Similarly, rs115550680 had interactions with two CpGs (cg06169110 and cg17316918). For those with the risk genotype (GG/AG), a 1% increase at cg06169110 was associated with a 0.37 SD decrease in general cognitive function (*p* = 2.00 × 10^−4^, Figure 2C), and a 1% increase at cg17316918 was associated with a 0.33 SD increase in general cognitive function (*p* = 0.004, Figure 2D).

We performed a sensitivity analysis by excluding outlying CpG values beyond four standard deviations of mean methylation, and our results remained consistent (Appendix A). To test whether the interaction was driven by potential SNP-CpG correlation, we assessed the association between each SNP-CpG pair. We observed nominal associations between rs3764647 and cg22271697, as well as between rs115550680 and cg06169110. For these two SNP-CpG pairs, we regressed out the SNP effect from the corresponding CpGs and re-tested the interactions. The results remained consistent with those reported in Table 3 (Appendix A). We also tested the association between all four significant interactions with each of the five neurocognitive domains. Similar interactions were observed for multiple neurocognitive measures, especially DSST and SCWT, in which all four interactions were significantly associated (Appendix A).

### 3.7. Gene Expression Associations

To understand the functional effects of identified SNP-by-CpG interactions, we examined their interaction effects (Appendix A) as well as marginal effects (Appendix A) on *ABCA7* gene and transcript expression. At the gene level, none of the identified SNP-by-CpG interactions were associated with gene expression in our sample. However, we found a negative association between one of the SNPs, rs115550680, and gene level expression of *ABCA7* (ENSG00000064687): for each additional rs115550680 G allele, there was a 0.05 decrease in gene expression (*p* = 0.027).

At the transcript level, two SNP-by-CpG interactions (rs115550680*cg17316918 and rs3764647*cg22271697) were nominally associated with two different transcripts (ENST00000525939 and ENST00000531467) (Appendix A). ENST00000531467 (Chromosome 19: 1,062,261–1,063,945 forward strand) is a protein coding transcript with four coding exons (Appendix A). ENST00000525939 (Chromosome 19: 1,062,261–1,063,945 forward strand) is a retained intron, found primarily in the spleen, pituitary, whole blood, and brain (cerebellum and cerebellar hemisphere) (Appendix A). Although the interactions were only nominally significant, we performed contrast tests to estimate the effect size of the CpG site in each genotype group on each identified transcript. Contrast tests showed that methylation at cg17316918 trended toward a positive association with ENST00000525939 among those with the rs115550680 risk genotype (GG/AG) but did not reach nominal significance (Appendix A). We also assessed the marginal associations of the two SNPs and two CpGs involved in the interactions on each of the *ABCA7* transcripts (Appendix A). We found that rs115550680 was negatively associated with 11 *ABCA7* transcripts, including ENST00000531467, at FDR q < 0.1 (Appendix A). Rs3764647 was nominally associated with only ENST00000530703 (*p* = 0.037; Appendix A). Among the CpGs involved in the interactions, cg06169110 was nominally associated with two transcripts (Appendix A).

The iMETHYL [62] *cis*-eQTM results for PBMCs and the three white blood cell types showed that there were CpGs within the *ABCA7* region, including within the promoter region, that regulate expression of both *ABCA7* and nearby genes. However, the CpGs identified in the significant SNP-by-CpG interactions in our study were not associated with gene expression of *ABCA7* or nearby genes at FDR q < 0.05.

## 4. Discussion

While previous studies have implied that *ABCA7* is a causal gene for AD [63,64,65,66], there is a dearth of studies examining the relationship between *ABCA7* and cognitive function. AD is a gradual neurodegenerative disease, characterized by noticeable cognitive impairment in areas of episodic memory, semantic memory, and executive function, with pathophysiology preceding the illness decades prior [67,68]. Studying the relationship between SNPs and CpGs in *ABCA7* and cognition may enhance our understanding of cognitive health and further elucidate the role of *ABCA7* in cognitive aging preceding AD. To our knowledge, this study is the first assessment of the association and interaction between DNA methylation and genetic risk factors in *ABCA7* on cognition in AA without dementia.

In this study, we found no association between known AD-associated SNPs and cognitive measures. This is, perhaps, not surprising, as previous studies have been inconsistent regarding the association between *ABCA7* SNPs and cognition. Most of the studies, however, have been conducted in primarily European ancestry populations [27,28,29,69]. For example, the Three-City Dijon study found no association between *ABCA7* common variants and global cognition, nor other cognitive outcomes [69]. Other studies in EA have shown that SNPs may be associated with cognition in subgroups stratified on gender [27], *APOE* status [28], or disease progression [29]. In light of this, we also assessed whether *ABCA7* SNP associations are modified by sex, *APOE* major isoforms, and/or education status. Unlike prior studies [27,28], we did not find any evidence of interaction. Lack of association with cognition for the sentinel SNP-by-sex and SNP-by-*APOE* interactions may be due to differences in ancestry or to small sample size, as those studies have sample sizes ranging from 1153 to 3267 [39,40]. Our study also did not find SNP-by-education associations interactions on cognition. This is consistent with another study, which observed no interaction between education and *ABCA7* variants on memory performance in either EA or AA; however, a weak signal was observed for memory decline in AA, which is a cognitive measure more closely related to AD and dementia than general cognitive function [70].

Other lines of evidence also suggest that the *ABCA7* risk variants may not be highly relevant to the neurological pathways underlying normal cognitive function and/or cognitive reserve. For example, previous GWAS for general cognitive function and AD have shown few overlapping loci [35,71]. Further, studies of cognitively “resilient” individuals who live to an older age with intact cognitive function, despite the presence of AD neuropathology, have found the genetic architecture of cognitive resilience to be distinct from that of AD [72]. At this point, relatively little is known about the pathways involving genetic variants and cognitive aging in those without dementia. Thus, studying variants that affect general cognitive function before development of dementia may identify novel pathways for therapeutic targets.

Only one epigenome-wide association study (EWAS) has examined the association between all CpG sites across the genome, including CpGs in *ABCA7* gene, and general cognitive function in participants from multi-ethnic backgrounds [73]. This study did not identify any significant associations between *ABCA7* and general cognitive function. However, due to the large numbers of CpG sites tested, the EWAS could have missed signals with smaller effect sizes. Moreover, the EWAS sample was mostly composed of EA. Our study, which focused on CpG sites in *ABCA7* in an AA cohort, would give us more power to detect an association in this region among AA. Nevertheless, we also failed to detect any associations between CpGs and general cognitive function after multiple testing correction, although six CpGs were associated at a nominal level. Importantly, we examined methylation levels in whole blood leukocytes, which is not the most relevant tissue for brain function. A study in post-mortem brain tissue found associations between CpGs in *ABCA7* and AD, as well as an increased burden of pathologies (e.g., Aβ load and tau tangle density), whereas another study failed to demonstrate differential methylation in peripheral blood between AD patients and controls [21]. Although methylation patterns differ between blood and brain tissues [23,74], blood cells touch every cell bed that affects the brain, and are related to chronic inflammation and oxidative stress, which are linked to cognitive performance [75,76]. Studying methylation in blood also allows us to study epigenetic associations with cognition in living participants in an inexpensive and non-invasive manner.

Although *ABCA7* sentinel SNPs and CpG sites were not associated with general cognitive function, we did find evidence of SNP-by-CpG interactions. Four interactions reached FDR significance (rs3764647*cg00135882, rs3764647*cg22271697, rs115550680*cg06169110, and rs115550680*cg17316918). Further, a total of nine CpG sites had at least nominally significant interactions with rs115550680 on cognition function. For participants who are homozygous for the rs115550680 major allele (AA), local methylation does not seem to have an effect on cognitive function. However, for participants who carry the risk allele (GG/AG), methylation at local CpGs may play an important role in cognition. This might be related to the different *ABCA7* transcripts that are involved in each case. Rs115550680 is located in an LD block that spans several introns and exons [8]. A prior study suggested that there is a 44-base pair exonic deletion (rs142076058, p.Arg578 fs) among rs115550680 G carriers, which could cause a frameshift in the *ABCA7*-coding sequence, resulting in the formation of a premature termination codon [77]. Indeed, our gene expression analysis found that the risk allele (G) at rs115550680 was strongly associated with decreased expression of 11 *ABCA7* transcripts. Taken together, these data suggest that this SNP might influence the major isoforms that are expressed, and the expressed alternative transcripts may influence cognitive function. Furthermore, alternative transcripts that are expressed in those carrying the risk allele may be further modulated by methylation level at local CpG sites, which may lead to differences in cognitive function in this group. Consistent with this hypothesis, methylation at cg17316918 was associated with transcript ENST00000525939 in rs115550680 risk allele carriers (GG/AG) only. Interestingly, this transcript is largely expressed in the brain. However, there is no prior evidence to show an association between this transcript and AD and/or cognition. Nonetheless, alternative splicing of *ABCA7* is likely to play a similar important role in cognition, as has been demonstrated in AD [78,79]. 

The other SNP that had significant interactions with *ABCA7* CpG sites, rs3764647, is a missense mutation, where the risk allele (G) leads to the amino acid change p.His395Arg in the first extracellular loop of the *ABCA7* protein [16]. One CpG site (cg00135882) is associated with cognitive function in participants who carry the risk allele (GG/AG), and another CpG site (cg22271697) is associated with cognitive function in those who do not carry the risk allele (AA). This differential pattern may be due to different functions of the two transcripts instead of alternative splicing. Consistently, we did not observe a direct association between this SNP or CpG with expression of *ABCA7* transcripts. Notably, three of the CpGs (cg00135882, cg22271697, and cg06169110) in the significant SNP-by-CpG interactions were either flanking or within CGIs. Active intragenic CGIs may change the major isoforms that are expressed by interfering with splicing and/or polyadenylation. Alternatively, they may promote enhancer function or act directly as an enhancer to regulate gene expression [19]. Consistent with this hypothesis, all four CpGs are located in regions that contain at least one important regulatory element (i.e., promoters, enhancers, and/or CTCF binding sites). Taken together, these results suggest that SNPs and CpG sites in *ABCA7* may interact to modulate the expression and/or function of *ABCA7* transcripts, and that some of the affected transcripts may influence cognitive function in older AA.

Indeed, recent literature suggests that SNP-by-CpG interactions might be an important mechanism underlying human complex diseases [80,81,82]. Similar SNP-by-CpG interactions have been identified in association with complex human disorders, such as breast cancer [83], type 2 diabetes [84], alcohol dependence [85], and suicide attempts in schizophrenia [86]. One factor to note, however, is that SNPs could have a cis-regulatory effect on local CpGs, which could cause a spurious interaction. However, our sensitivity analysis demonstrated that the interactions which we observed were not solely due to SNP-CpG correlations. In summary, we have demonstrated that a complicated interplay between genetic and epigenetic risk factors in the *ABCA7* region may play an important role in cognitive function. Future studies are needed to disentangle this complicated relationship.

Our study is not without limitations. First, our gene expression measures were taken from transformed B-lymphocytes from immortalized cell lines. While transformed B-lymphocytes are a convenient source of DNA, the transformation process causes epigenetic changes to the immortalized cells that are not fully understood [87]. However, they provide a unique and efficient way to examine the functional effects of genetic and epigenetic variation on gene expression, since the environmental conditions of the cells are the same across individuals. In addition, previous *cis*-eQTM studies in white blood cells have shown that at least some CpGs within the *ABCA7* region promote or repress gene expression of *ABCA7* and nearby genes, but we did not observe eQTM relationships with those same CpGs in our study. One reason for this may be that our methylation was measured in blood and included a mix of white blood cells, while our gene expression was measured in transformed B-lymphocytes. Additional work is needed to understand how *ABCA7* CpGs and their interactions with SNPs influence proximal gene expression in a variety of white blood cell types, which would further shed light on the complicated biological mechanisms that contribute to cognitive function. We also acknowledge that our findings need to be replicated in a larger sample of AA. Further studies in animal and cellular models are also warranted to confirm our findings and to reveal how SNPs and methylation jointly contribute to cognitive function. Finally, due to the cross-sectional nature of our study, we cannot infer causality of our findings or quantify how the SNP-by-CpG interactions alone impact cognition. To that end, longitudinal studies are necessary to investigate how SNPs and/or CpGs affect cognitive changes over time. 

Our study also has notable strengths. To our knowledge, our study is the first to take a multi-omic approach to investigate the relations between the *ABCA7* gene region and cognitive function in a population-based cohort of older adults without diagnosed dementia. Our study was also conducted on AA, an understudied population with a higher prevalence of AD [3,5] and higher conferred risk of AD from *ABCA7,* compared to EA [8]. Additionally, with comprehensive cognition measures, we were able to assess associations with multiple neurocognitive domains, as well as general cognitive function.

## 5. Conclusions

In the present study, we evaluated the association between *ABCA7* genetic, epigenetic, and transcriptomic markers and cognitive function in 634 AA participants without preliminary evidence of dementia. We found that DNA methylation levels at local CpG sites modify the relationship between genetic variants and general cognitive function. Specifically, two SNPs in the *ABCA7* gene region (rs3764647 and rs115550680) may regulate the effects of methylation on cognition. Differential gene expression analysis further highlighted the potentially causal transcripts. In conclusion, our findings suggest that a complicated interplay between genetic and epigenetic factors in *ABCA7* may influence cognition in older AA without dementia.

## Figures and Tables

**Figure 1 genes-13-02150-f001:**
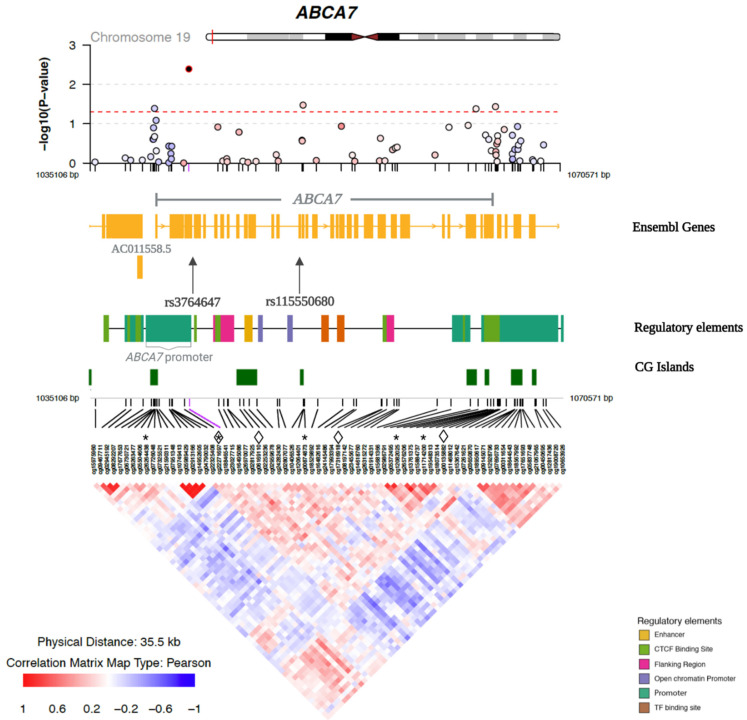
Regional plot of the association between DNA methylation in the *ABCA7* region and general cognitive function. The top panel shows −log10 (*p* value) for the association between methylation and general cognitive function, adjusting for age, sex, age difference between methylation and cognition measurements, educational attainment, *APOE* ε2, *APOE* ε4, smoking status, PC1-4, and familial relatedness (random effects; Model 3), according to chromosomal positions. Nominally significant (*p* < 0.05) associations are above the dashed line. The middle panels show Ensembl genes, regulatory elements, and CpG islands (UCSC Genome Browser) in the *ABCA7* region. The lower panel shows the correlations in the DNA methylation levels among the 72 CpG sites in this region. The five CpGs that have a nominal association with general cognitive function are marked by asterisks. The four CpGs and two SNPs that were identified in the SNP-by-CpG interactions associated with general cognitive function are marked by diamond symbols (CpGs) and arrows (SNPs).

**Figure 2 genes-13-02150-f002:**
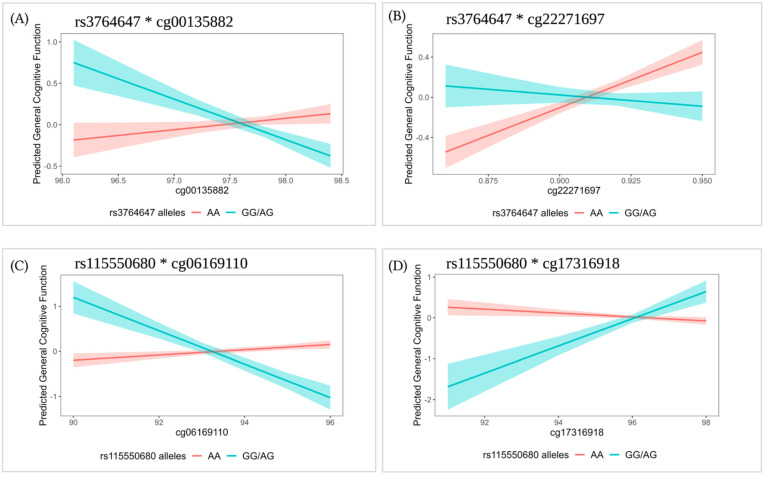
Linear prediction of CpG sites (% methylated) on general cognitive function for a given SNP genotype group in the *ABCA7* region: (**A**) rs3764647*cg00135882, (**B**) rs3764647*cg22271697, (**C**) rs115550680*cg06169110, and (**D**) rs115550680*cg17316918. Models were adjusted for age, sex, age difference between methylation measurement and cognition measurement, educational attainment, *APOE* ε2, *APOE* ε4, smoking status, PC1-4, and familial relatedness as a random effect (Model 4). Regression lines are shown with standard error bands. For rs3764647, GG (*n* = 17) and AG (*n* = 156) groups were combined in the GG/AG group (*n* = 173). For rs115550680, GG (*n* = 5) and AG (*n* = 54) groups were combined in the GG/AG group (*n* = 59).

**Table 1 genes-13-02150-t001:** Sample characteristics of Genetic Epidemiology Network of Arteriopathy (GENOA) African Americans (*n* = 634).

	Mean (SD) or *n*%
Age at cognition measurement (years)	63.31 (8.08)
Age difference between methylation and cognition measurements (years) ^a^	6.03 (1.29)
Sex	
Female	475 (74.90%)
Male	159 (25.10%)
Educational attainment	
At least some college	300 (47.32%)
High school degree/GED	169 (26.66%)
Less than high School degree/GED	165 (26.03%)
Smoking status	
Current smoker	105 (16.56%)
Former smoker	146 (23.03%)
Never smoker	383 (60.41%)
General cognitive function	0.00 (1.00)
Delayed recall (RAVLT, number of words recalled)	7.05 (3.34)
Processing speed (DSST, number of symbols)	34.44 (12.62)
Word fluency (COWA-FAS, number of words)	29.73 (11.61)
Concentration effectiveness (SCWT, number of items)	22.53 (9.83)
Visual conceptual tracking (TMTA, seconds to test completion)	61.63 (31.96)

Abbreviations: HS, high school; RAVLT, Rey Auditory Verbal Learning Test; DSST, Digit Symbol Substitution Test; COWA-FAS, Controlled Oral Word Association Test; SCWT, Stroop Color–Word Test; TMTA, Trail Making Test A. ^a^. The subset sample (*n* = 494) consists of subjects with available genetic and methylation data.

**Table 2 genes-13-02150-t002:** Interaction between *ABCA7* sentinel SNPs and CpG sites on general cognitive function (FDR q < 0.1; *n* = 494).

	SNP Annotation				CpG Site Annotation				Main Effects				Interaction	
SNP*CpG Site Interaction	SNP	Position	Risk Allele	RAF	Cpg Site	Position	Site Type	Relation to CGI	β_SNP_	*p*-Value	β_CpG_	*p*-Value	β_interaction_	*p*-Value
rs3764647*cg00135882	rs3764647	1044712	G	0.20	cg00135882	1065783	Promoter	North Shore	−0.01	0.875	0.24	0.086	−0.80	1.46 × 10^−4^ **
rs3764647*cg22271697	rs3764647	1044712	G	0.20	cg22271697	1042537	Promoter	North Shelf	−0.07	0.319	0.16	7.23 × 10^−6^ *	−0.18	5.77 × 10^−4^ **
rs115550680*cg06169110	rs115550680	1050420	G	0.06	cg06169110	1046615	Gene Body	CG Island	−0.23	0.045 *	0.06	0.143	−0.38	2.18 × 10^−4^ **
rs115550680*cg17316918	rs115550680	1050420	G	0.06	cg17316918	1056930	Gene Body	Open Sea	−0.05	0.661	−0.06	0.164	0.41	4.84 × 10^−4^ **

Abbreviations: AA, African American; EA, European American; RAF, risk allele frequency in GENOA. Model 4: General cognitive function ~ SNP + CpG + SNP*CpG + age at cognitive testing + age difference between methylation and cognition measurements + sex + educational attainment + APOE ε2 + APOE ε4 + smoking status + PC1-4 + familial relatedness (random effect). * *p* < 0.05, ** FDR q < 0.1.

**Table 3 genes-13-02150-t003:** Estimated effect of CpG site on general cognitive function for the given *ABCA7* SNP genotype group (*n* = 494).

SNP	Cpg Site	Genotype	β_CpG_	*p*-Value
rs3764647 ^a^	cg00135882	AA	0.09	0.566
		GG/AG	−0.68	0.004 *
rs3764647 ^a^	cg22271697	AA	0.14	2.00 × 10^−4^ *
		GG/AG	−0.02	0.719
rs115550680 ^b^	cg06169110	AA	0.05	0.221
		GG/AG	−0.37	2.00 × 10^−4^ *
rs115550680 ^b^	cg17316918	AA	−0.06	0.202
		GG/AG	0.33	0.004 *

^a^: GG (*n* = 17) and AG (*n* = 156) groups were combined in the GG/AG group (*n* = 173). ^b^: GG (*n* = 5) and AG (*n* = 54) groups were combined in the GG/AG group (*n* = 59). Model 4: General cognitive function~ SNP + CpG + SNP*CpG + age at cognition measurement + age difference between methylation and cognition measurements + sex + educational status + *APOE* ε2 *+ APOE* ε4 + smoking status + PC1-4 + familial relatedness (random effect). * *p* < 0.05.

## Data Availability

The phenotype data and *APOE* genotypes used in the current study are available upon reasonable request to J.A.S. and S.L.R.K., and with a completed data use agreement (DUA). All other genotype data are available from the Database of Genotypes and Phenotypes (dbGaP): phs001401.v2.p1. Methylation and gene expression data are available from the Gene Expression Omnibus (GEO): GSE210256 and GSE138914. Due to IRB restriction, mapping of the sample IDs between genotype data (dbGaP) and methylation data (GEO) cannot be provided publicly, but is available upon written request to J.A.S. and S.L.R.K.

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
