# Peer review of "SNP-by-CpG Site Interactions in ABCA7 Are Associated with Cognition in Older African Americans"

_genes, 2022, doi:10.3390/genes13112150_

Round 1
Reviewer 1 Report
Remarks:
Dima Chaar et al. submitted their manuscript “SNP-by-CpG site interactions in ABCA7 are associated with cognition in older African Americans” for publication to Genes. Here, they have made associations between the known SNPs and DNA methylation levels on the gene ABCA7 to see how the interaction between the SNP-Epigenetic modifications have an effect on the cognitive abilities of older African American populations. They have found that DNA methylation levels at local CpG sites modify the relationships between genetic variants and general cognitive function. In particular, 2 SNPs in the ABCA7 gene region seem to show regulatory effects of methylation on cognition. Their findings suggest that there is a complicated interplay between genetic and epigenetic factors in this gene ABCA7 may influence cognition in older AA without dementia. However, there are major issues that needs to be addressed in their manuscript:
Major Issues:
-
In Figure 1, they need to label the significant SNPs they found to have regulatory effects. Highlighting them would be beneficial to understand the diagram better.
-
Using publicly available myeloid specific transcriptomic data, the authors can examine whether the regulatory elements, especially the promoter regions within the ABCA7 gene as shown in Figure 1, is regulating the ABCA7 gene itself or other nearby genes.
-
They need to provide sufficient information regarding why they are even looking into the CpG sites in the first place. It needs some hypothesis and rationale. What makes CpG island important to this gene. The authors could provide more clear evidence in the text.
-
Intragenic CpG are known to be silenced, impact pre-mRNA processing and promote or contain enhancer function (doi: 10.3389/fcell.2022.832348). The authors could infer a mechanistic insight of CpG regulation to their findings of ABCA7 gene.
-
They did not particularly mention the mechanistic understanding of why SNP and DNA methylation on the ABCA7 gene affected cognition. Their finding suggests a complicated interplay which influences the outcome of the cognition, so we cannot say how much impact does SNP-DNA methylation solely has on the cognitive outcome.
-
The paper’s flow of information needs more improvement. For example, the overview flow chart was at the end of the supplementary information. This flow chart needs to be at the beginning of the supplementary information or included in the main text. Same goes for the transcript figures from the GTEx tissue data. This plot needs to be mentioned and shown at the earlier stages of the paper, so this information can be clearly mentioned.
-
It is recommended to include a figure or a flowchart of the 4 MODELS they used to normalize their SNP-CpG interaction datasets. They have only been mentioned in the text or legend.
-
It will be helpful to provide eQTL results of ABCA7 for each of the 4 SNP-CpG plots made in figure 2. This will show if the cognitive changes we see in those plots are also related to changes in gene expression levels of ABCA7.
Minor Issue:
-
The text in the discussion is not consistent. Please use the same text size and font.
-
For transcript information, they should illustrate the respective transcripts affected by the 2 major SNPs and CpG island methylation sites in the main text. They should annotate the exons and the intragenic regions of those transcripts much more clearly and generate a figure as part of the main text.
Author Response
Please see the attachment for the response to Reviewer 1's comments. Thank you.

Reviewer 2 Report
SNPs in ABCA7 confer the largest genetic risk for Alzheimer’s Disease (AD) in African Americans (AA) after APOE ε4. However, the relationship between ABCA7 and cognitive function has not been thoroughly examined. We investigated the effects of 5 known AD risk SNPs and 72 CpGs in ABCA7, as well as their interactions, on general cognitive function (cognition) in 634 older AA without dementia from Genetic Epidemiology Network of Arteriopathy (GENOA). Using linear mixed models, no SNP or CpG was associated with cognition at FDR q<0.1, but 5 CpGs were nominally associated (P<0.05). Four SNP-by-CpG interactions were associated with cognition (FDR q<0.1).Blood samples were genotyped using the Affymetrix® Genome-Wide Human SNP. Conclusins, In the present study, we evaluated the association between ABCA7 genetic, epigenetic,
and transcriptomic markers and cognitive function in 634 AA participants without
preliminary evidence of dementia. We found that DNA methylation levels at local CpG
sites modify the relationship between genetic variants and general cognitive function.
Specifically, two SNPs in the ABCA7 gene region (rs3764647 and rs115550680) may regulate
the effects of methylation on cognition. Differential gene expression analysis further
highlighted the potentially causal transcripts. In conclusion, our findings suggest that a
complicated interplay between genetic and epigenetic factors in ABCA7 may influence
cognition in older AA without dementia.
Array 6.0 or the Illumina 1M Duo. Samples they also applied methylation measures and computed correlations + applied epigenetic tests there are 2 figures and no major spelling errors thus the paper can be accepted as is for publication
And find herein
1. The paper is of interest and very relevant. The data was collected from patient's samples (but the ethics statement was missing pls ask the authors to add it if they have one , and they can also add the BRAAK stage for the patients if known) 2. The data was made available under the GEO. 3. They mention the study limitations which is appropriate. 4. The figure is of high resolution 5. They appalled FDR in their statistics. 6. The authors can add reference to the following paper-The Amyloid-β Pathway in Alzheimer's Disease.
Author Response
Please see the attachment for the response to Reviewer 2's comments. Thank you.
